# Diversity and Interactions between Picobiine Mites and Starlings

**DOI:** 10.3390/ani14172517

**Published:** 2024-08-29

**Authors:** Bozena Sikora, Jakub Z. Kosicki, Milena Patan, Iva Marcisova, Martin Hromada, Maciej Skoracki

**Affiliations:** 1Department of Animal Morphology, Faculty of Biology, Adam Mickiewicz University, Uniwersytetu Poznańskiego 6, 61-614 Poznań, Poland; boszka@amu.edu.pl (B.S.); milpat@st.amu.edu.pl (M.P.); 2Department of Avian Biology and Ecology, Faculty of Biology, Adam Mickiewicz University, Uniwersytetu Poznańskiego 6, 61-614 Poznań, Poland; kubako@amu.edu.pl; 3Laboratory and Museum of Evolutionary Ecology, Department of Ecology, Faculty of Humanities and Natural Sciences, University of Prešov, 08001 Prešov, Slovakia; marcisova.iva@gmail.com (I.M.); hromada.martin@gmail.com (M.H.)

**Keywords:** Acariformes, Aves, birds, ectoparasites, parasites, Picobiinae, quill mites, Sturnidae, Syringophilidae

## Abstract

**Simple Summary:**

This study investigates the complex interactions between quill mites of the family Syringophilidae and starlings. We identified five species of quill mites infesting 24 species of starlings, uncovering intricate host–parasite dynamics across the Palaearctic, Afrotropical, Oriental, and Oceanian zoogeographical regions. A detailed statistical analysis of the Picobiinae–Sturnidae network revealed low connectivity but high modularity, indicating specific relationships between the mites and their starling hosts. The network structure demonstrated four distinct modules, highlighting the specialised and compartmentalised characteristics of these associations. Furthermore, the distribution of *Picobia* mites was found to align with the phylogeny of their starling hosts, with particular mites targeting specific starling clades. The social and breeding behaviours of starlings were observed to contribute to the high prevalence of these mites. This comprehensive network analysis provides new insights into the ecological dynamics of host–parasite interactions.

**Abstract:**

The subfamily Picobiinae (Acariformes: Syringophilidae) comprises obligate and permanent parasites of birds found exclusively in the quills of contour feathers. We studied associations of picobiine mites with birds of the family Sturnidae (Aves: Passeriformes) across the Palaearctic, Afrotropical, Oriental, and Oceanian zoogeographical regions. Among the 414 examined bird individuals belonging to 44 species (35.2% of all sturnids), 103 individuals from 24 species (54.5% of examined species) were parasitised by quill mites. The diversity of mites was represented by five species, including one newly described, *Picobia malayi* Patan and Skoracki sp. n. Statistical analysis of the Picobiinae–Sturnidae bipartite network demonstrated a low connectance value (Con = 0.20) and high modularity, with significant differences in the H_2_′ specialisation index compared to null model values. The network structure, characterised by four distinct modules, highlighted the specificity and limited host range of the Picobiinae–Sturnidae associations. The distribution of *Picobia* species among starlings was congruent with the phylogeny of their hosts, with different mites parasitising specific clades of starlings. Additionally, the findings suggest that the social and breeding behaviours of starlings influence quite a high prevalence. Finally, our studies support the validity of museum collections to study these parasitic interactions.

## 1. Introduction

The subfamily Picobiinae, established by Johnston and Kethley in 1973, comprises obligate and permanent parasites of birds, exclusively found in the quills of contour feathers covering the head, neck, breast, abdomen, and back of their avian hosts [1,2]. An exception within this subfamily is *Calamincola lobatus* Casto, 1977, a unique species belonging to a monotypic genus, which inhabits the quills of wing feathers of the Groove-billed Ani, *Crotophaga sulcirostris* Swainson (Cuculiformes: Cuculiudae) [3]. Within the quills, picobiines feed on soft tissue fluid by piercing the quill wall with their long needle-like chelicerae [4,5]. The entire life cycle of these mites occurs within the quills, from larva through two nymphal stages (protonymph and deutonymph) to adults. The dispersal stage involves adult and fertilised females, which seek new feathers to infest, either on the same host individual or through vertical transmission from parent to chick [4,6,7].

Picobiines form associations exclusively with neognathous birds (Neognathae), spreading across 10 different orders [8,9]. Each avian order typically hosts a unique picobiine genus. For example, birds of the order Columbiformes are infested by *Gunabopicobia*, Galliformes by *Columbiphilus*, Cuculiformes by *Calamincola*, and Charadriiformes by *Charadriineopicobia*. In contrast, other bird orders host multiple picobiine genera. Among bird orders, the most diverse picobiine mite fauna is found in Passeriformes, the most species-rich avian order, which is infested by representatives of five genera: *Neopicobia*, *Picobia*, *Phipicobia*, *Pipicobia*, and *Rafapicobia* [10,11,12]. Similarly, the order Psittaciformes hosts members of the four genera: *Lawrencipicobia*, *Rafapicobia*, *Neopicobia*, and *Pipicobia* [13,14]. This highlights the great diversity within this parasitic mite subfamily.

Since the early 2000s, research has significantly expanded our understanding of picobiine diversity, with numerous new species and genera being described. Comprehensive reviews of picobiine fauna have been conducted across various zoogeographic regions, covering all except Antarctica (see references in [8]). Currently, the subfamily Picobiinae comprises 116 species grouped into 12 genera [8,9,15]. Due to the vast diversity of these mites, their wide host range, and the many avian hosts yet to be studied, current research represents only the beginning of uncovering the full extent of picobiine fauna.

The largest avian order, Passeriformes, comprises more than 6200 extant species grouped into over 1350 genera and 140 families [16]. The research presented here continues large-scale studies focusing on quill mites across particular families within the order Passeriformes.

The family Sturnidae (Starlings) includes approximately 125 species grouped into 36 genera [17,18]. Starlings are medium-sized birds that adapt to various habitats, including dry savannas, scrublands, farmlands, grasslands, urban areas, and forests in temperate and tropical climates [17]. These birds are found in the Palaearctic, Afrotropical, Oriental, and Oceanian zoogeographical regions [16]. Starlings are social birds with often nomadic habits, assembling in huge flocks. Some species breed in colonies, which can be very large (even tens of thousands of pairs). They mostly breed in cavities, but some species also place their nests in the open. Their evolutionary success can be attributed to their high degree of dispersal. Cooperative breeding is also found in many sturnid species [16]. All these traits make starlings very interesting as hosts of highly specific quill mites, including picobiines. Unfortunately, our understanding of quill mites parasitising this avian family remains incomplete. To date, three species of the genus *Picobia* Heller, 1880, have been described from this host group [19,20,21].

The current study has the following objectives: first, it describes a new species of the genus *Picobia*, and second, it aims to detail the diversity, interactions, and level of specialisation between syringophilid ectoparasites and their starling hosts on a worldwide scale.

## 2. Materials and Methods

### 2.1. Host Sampling

The mite material used in this paper was collected from dry bird skins primarily housed in the ornithological collection of the Bavarian State Collection of Zoology (SNSB-ZSM) in Munich, Germany. Additionally, the Eurasian Starling (*Sturnus vulgaris*) was examined at the Icelandic Institute of Natural History (IINH) in Reykjavik, Iceland, and the Superb Starling (*Lamprotornis superbus*) was examined in the Ornithology Section of the National Museum of Kenya (NMK) in Nairobi, Kenya. Approximately 10 contour feathers were removed from each bird specimen, specifically from the area near the cloaca. 

### 2.2. Mites Preparation, Identification, and Depository

All collected feathers were examined under a stereomicroscope and opened using fine forceps. Infested feathers were placed in tubes with Nesbitt’s solution at room temperature for 3 days, following the protocol introduced by Walter and Krantz [22] and Skoracki [11]. Subsequently, mites were mounted on microscope slides in Hoyer’s medium [22]. Identifications and drawings of mite specimens were conducted using an Olympus BX51 light microscope (Olympus Corp., Tokyo, Japan) with differential interference contrast (DIC) optics. Figures were prepared using a camera lucida attachment. In the descriptions, all measurements are provided in micrometers; the dimension ranges of the paratypes are provided in parentheses, following the data from the holotype. The idiosomal setation follows Grandjean [23] as adapted for Prostigmata by Kethley [24]. The nomenclature of leg chaetotaxy follows that proposed by Grandjean [25]. The morphological terminology follows Kethley [1] and Skoracki et al. [8]. The abbreviations PF and NPF are used for the physogastric and non-physogastric forms of the females, respectively.

Mite specimens are curated at the following repositories, abbreviated as AMU for the Department of Animal Morphology, Adam Mickiewicz University in Poznań, Poland, and SNSB-ZSM for the Section Arthropoda Varia, Bavarian State Collection for Zoology, Munich, Germany.

The list with full data of the collected mite material is presented in Appendix A.

### 2.3. Bird Systematics and Zoogeographical Regions

The scientific names and systematics of the birds follow Clements et al. [18] and Winkler et al. [17]. Zoogeographical regions follow Holt et al. [26] and Ficetola et al. [27].

### 2.4. Statistical Analyses

To describe patterns within the studied host–parasite ecological two-way web, the “bipartite” package for R was used [28]. This approach allows for a quantitative description of the ecological connections between parasites and their hosts [29]. In our matrix, parasite prevalence was utilised as a quantitative index. Descriptive statistics were computed utilising Quantitative Parasitology v.3.0 on the Web [30,31,32].

First, we calculated connectance (connection ratio) in a bipartite network. This measure determines the ratio of the number of actual connections in the netDwork to the maximum possible number of connections. It indicates network density and is expressed as a value between 0 and 1, where 1 means that every possible element from one part of the network (parasites) is connected to every element from the other part of the network (hosts) and 0 means that there are no connections. Next, we calculated the C.score, a measure describing the tendency towards non-co-occurrence of species pairs. A high value of this index indicates a tendency for species to not co-occur, suggesting competition or other forms of mutual avoidance. Conversely, a low C.score indicates that species tend to co-occur more often than expected based on random distribution. To determine whether parasites are generalists or specialists, we used the H_2_′ measure. This statistic ranges from 0 to 1, where 0 indicates that all species are generalists (interacting with many hosts), and 1 indicates that all species are specialists (each species only interacts with one host). The null.t.test was employed to check whether the observed H_2_′ values significantly deviated from random values [33]. We also calculated nestedness, which is a measure of order in the network. If parasites with a low number of interactions with hosts share these interactions with other hosts with a larger number of interactions, the network is considered nested. Nestedness temperature measures how much the actual nestedness of a network differs from a perfectly nested network. For a perfectly nested network, the temperature would be 0. A higher temperature indicates less perfect nesting. As a measure of the functional diversity of parasites, we used d′ and d. The d′ is normalised to the maximum possible diversity for a particular species in the network, ranging from 0 to 1, with 1 indicating maximum functional diversity. If d′ for a species is 1, it means that it has maximum functional diversity, indicating that its interactions with other species are very diverse and unique compared to other species in the network. The d is a raw measure of functional diversity based on functional distances between species. High d values suggest a greater diversity of parasites in their interactions with hosts. Finally, we calculated modularity as a probability value that measures how well the modular structure fits the data. A value close to 1 indicates that the modular model has a relatively good fit for the observed data, suggesting that modularity is a significant feature of this network.

## 3. Results

### 3.1. Systematics

In our study, we identified five mite species of the subfamily Picobiinae parasitising 24 starling species (Table 1). Below, we present a list of the mite species along with their hosts and distribution, including new records. At the end of this section, we have constructed a key to all picobiine species associated with birds of the family Sturnidae.

#### 3.1.1. *Picobia malayi* Patan and Skoracki sp. n. (Figure 1 and Figure 2)

Female. Gnathosoma. Hypostomal apex with pair of blunt-ended projections. Peritremes V-shaped, each medial branch with seven chambers, each lateral branch with seven or eight clearly visible and striated chambers. Stylophore 300 (295–310) long, exposed portion of stylophore apunctate, 180 (175–190) long. Movable cheliceral digit 260 (260–270) long. Idiosoma. Propodontal shield divided into two narrow lateral sclerites and triangular medial sclerite; lateral sclerites punctate between bases of setae *ve* and *si*, bearing bases of setae *ve*, *si* and *se*, medial sparsely punctate. Setae *vi* and *ve* situated on same transverse level. Setae *c1* situated slightly anterior to level of setae *se* or both pair of setae set on the same level. Propodontal setae *vi*, *ve* and *si* beaded. Pygidial shield well developed and large in size, apunctate. Setae *h1* approximately 5.5 times longer than *f1*; setae *f2* about twice as long as *f1*. Bases of agenital setae *ag2* situated postero-lateral to setae *ag1*. Genital plate present, apunctate. Genital lobes absent. Pseudanal setae *ps1* and *ps2* subequal in length. All coxal fields well developed and apunctate. Legs. Antaxial and paraxial members of claws equal in size and shape. Lengths of setae: *vi* (170–195), *ve* 160 (130), *si* (200–230), *se* 310 (270–305) *c1* 290 (255–295), *c2* 305 (275–315), *d1* 165 (140–160), *d2* 290 (275–305), *e2* 180 (170–175), *f1* 65 (50–60), *f2* 110 (100–130), *h1* 355 (310–375), *h2* (590–610), *ag1* 95 (75–100), *ag2* 35 (30–40), *ag3* 180 (170–200), *g1* 30 (30–40), *ps1* 25 (25–30), *ps2* 25 (20–30), *l′RIII* 35 (30), *l′RIV* 30 (25–30), *3b* 40 (30), *3c* 150 (115–140), *4b* 35 (35–45), *4c* 150 (145–165).

Male. Not found. Figure 1*Picobia malayi* Patan and Skoracki sp. n., female. (**A**)—dorsal view; (**B**)—ventral view.
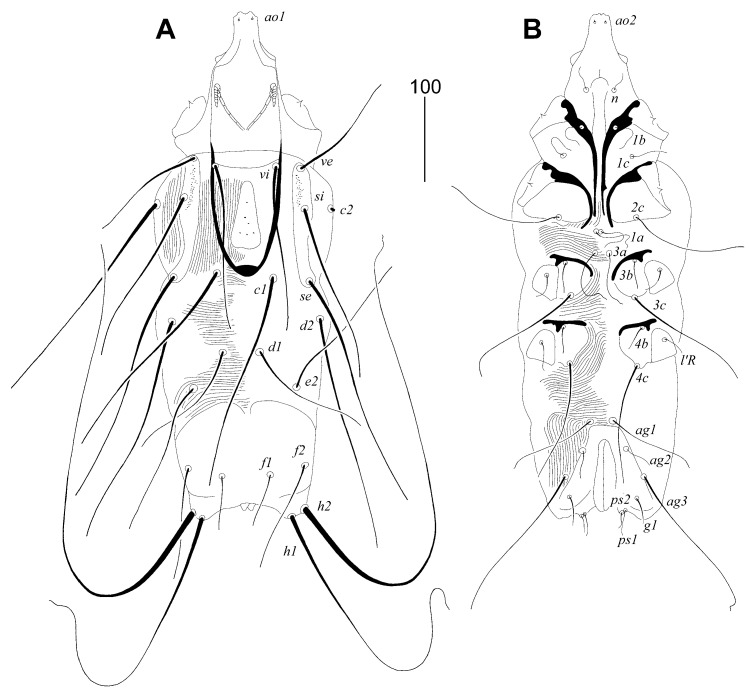

Figure 2*Picobia malayi* Patan and Skoracki sp. n., female. (**A**)—hypostomal apex; (**B**)—peritreme; (**C**)—propodonotal seta *si*; (**D**)—genito-anal region in ventral view; (**E**)—tarsus III with claws.
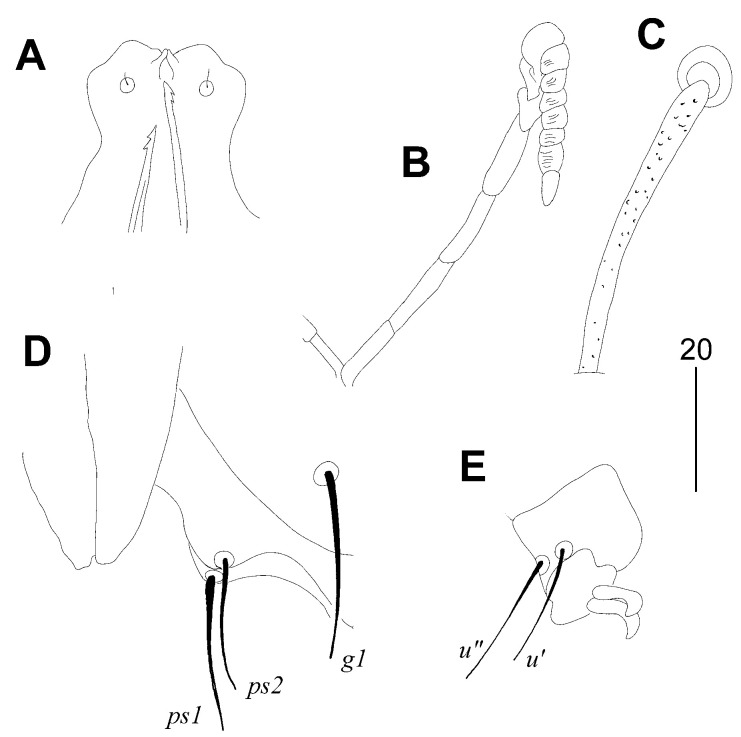


##### Type Material

Female holotype (NPF) and paratypes: six females (PF) and two females (NPF) from three specimens of the Common Hill Myna, *Gracula religiosa* Linnaeus, Indonesia, Malay Archipelago, Sumatra, 1908 (host specimens at SNSB-ZSM, uncatalogued). Mites removed by B. Sikora and M. Skoracki.

##### Type Material Deposition

Holotype and most of paratypes deposited in the AMU (reg. no. AMU MS 21-0910-059, 061, 089), except one female (PF) and one female (NPF) paratypes in the SNSB-ZSM.

##### Additional Material

Six females (NPF) and seven females (PF) (reg. no. AMU MS 21-0910-052) from the White-necked Myna, *Streptocitta albicollis* (Vieillot) (host at SNSB-ZSM, uncatalogued), Indonesia, Malay Archipelago, Celebes Isl., 1875, coll. Riedel. One female (NPF) and one female (PF) (reg no. AMU MS 21-0910-054) from the Coleto, *Sarcops calvus* (Linnaeus) (host reg. no. SNSB-ZSM 26.213), Philippines, Cebu Island, 1879, coll. Burger. One female (PF) (reg no. AMU MS 21-0910-056) from the same host species (host reg. no. SNSB-ZSM 26.213), Philippines, Luzon Island, Manila, 1963, coll. I. Marschdii (host at SNSB-ZSM, uncatalogued).

##### Differential Diagnosis

*Picobia malayi* sp. n. is morphologically similar to *Picobia indonesiana* Skoracki and Glowska, 2008. In females of both species, the hypostomal apex is with a pair of blunt-ended projections; each lateral branch of the peritremes has a similar number (seven or eight) of chambers; the propodonotal shield is divided into three sclerites (two lateral and unpaired medial); setae *h2* are distinctly longer than *f1*; the bases of setae *ag2* are situated postero-lateral to *ag1*. This new species is easily distinguishable from *P. indonesiana* by the following features: in females of *P. malayi*, each medial branch of the peritremes has seven chambers; the stylophore is 295–310 long; setae *h1* are 5.5 times longer than *f1*. In females of *P. indonesiana*, each medial branch of the peritremes has four or five chambers; the stylophore is 205 long; setae *h1* are about twice as long as *f1*.

##### Etymology

The specific epithet “malayi” originates from the Malay Archipelago, indicating the geographic region where the birds that act as hosts for this parasite species are found.

#### 3.1.2. *Picobia indonesiana* Skoracki and Glowska, 2008

Hosts and distribution. The Singing Starling *Aplonis cantoroides* (Gray) from Indonesia (Misool Isl.), the Long-tailed Starling *Aplonis magna* (Schlegel) from Indonesia (Numfor Isl.), the Metallic Starling *Aplonis metallica* (Temminck) from Papua New Guinea and Indonesia (Misool Isl.), the Asian Glossy Starling *Aplonis panayensis* (Scopoli) from Indonesia (Sangir, Sumatra and Java) and India [34] (current study), the Fiery-browed Starling *Enodes erythrophris* (Temminck) from Indonesia [34], the Golden Myna *Mino anais* (Lesson) from Papua New Guinea (current study), and the Yellow-faced Myna *Mino dumontii* (Lesson) from Indonesia [34] and Papua New Guinea (current study).

#### 3.1.3. *Picobia lamprotornis* Klimovicova, Skoracki, Wamiti and Hromada, 2014

Hosts and distribution. The Greater Blue-eared Glossy Starling *Lamprotornis chalybaeus* Hemprich and Ehrenberg from Tanzania and Kenya, the Lesser Blue-eared Glossy-Starling *Lamprotornis chloropterus* Swainson from Tanzania [21], the Superb Starling *Lamprotornis superbus* Rüppell from Kenya [19,21], the Abbott’s Starling *Poeoptera femoralis* (Richmond) from Tanzania, and the Kenrick’s Starling *Poeoptera kenricki* Shelley from Tanzania [35].

#### 3.1.4. *Picobia sturni* Skoracki, Bochkov and Wauthy, 2004

Hosts and distribution. The Crested Myna *Acridotheres cristatellus* (Linnaeus) from China, the Jungle Myna *Acridotheres fuscus* (Wagler) from Nepal, the Common Myna *Acridotheres tristis* (Linnaeus) from India (current study), the Wattled Starling *Creatophora cinerea* (Meuschen) from Kenya and Tanzania [35] (current study), the White-cheeked Starling *Spodiopsar cineraceus* (Temminck) from Japan and China [11] (current study), the Red-billed Starling *Spodiopsar sericeus* (Gmelin) from China (current study), the Spotless Starling *Sturnus unicolor* Temminck from Morocco [8], Italy and Spain (current study), the Eurasian Starling *Sturnus vulgaris* Linnaeus from Poland, Moldova [20,36], Slovakia [11], Iceland, Kazakhstan, Kyrgyzstan, Uzbekistan, and China (current study).

#### 3.1.5. *Picobia wisniewskii* Patan, Skoracki and Marcisova, 2024

Hosts and distribution. The Red-winged Starling *Onychognathus morio* (Linnaeus) from Tanzania [35] and the Bristle-crowned Starling *Onychognathus salvadorii* Sharpe from Ethiopia (current study).

#### 3.1.6. Key to the Species of the Picobiine Mites (Females) Associated with Starlings

1. Opisthonotal setae *h1* distinctly longer than *f1*. Genital lobes absent … 2

–Opisthonotal setae *h1* and *f1* subequal in length. Genital lobes present … 3

2. Setae *h1* 5.5 times longer than *f1* … *P. malayi* sp. n.

–Setae *h1* twice as long as *f1* … *P. indonesiana*

3. Genital lobes weakly developed and blunt-ended … 4

–Genital lobes well-developed and sharp-ended … *P. sturni*

4. Medial sclerite of propodonotal shield is punctate. Length of stylophore 235–240.

Lengths of setae *c2* and *d2* 230–240 and 230–240, respectively … *P. wisniewskii*

–Medial sclerite of propodonotal shield is apunctate. Length of stylophore 195–200.

Lengths of setae *c2* and *d2* 175–180 and 165–190, respectively … *P. lamprotornis*

### 3.2. Prevalence

Among the 414 examined bird individuals belonging to 44 species (35.2% of all sturnid species), 103 individuals from 24 species (54.5% of examined species) were parasitised by quill mites (Table 2). The species richness of quill mites associated with Starlings is represented by five species, including a new species described above. The prevalence varied considerably, ranging from 5.9% to 100% for particular avian species, and the confidence intervals were wide, ranging from 0.3 to 100. The examined avian material included 11 uninfested host species (N = 59) (Table 3).

### 3.3. Statistical Analysis

The Picobiinae–Sturnidae bipartite network (Figure 3) exhibited a low connectance value (Con = 0.20). This result is also confirmed by a C-score = 1, H_2_′ specialisation index = 1. A comparison between H_2_′ and null model values revealed significant differences (mean H_2_′ for null model = 0.028, t = −678.68, *p* < 0.0001). The temperature of nestedness = 44.45. The normalised specialisation level (d′) (maximum functional diversity) in all cases = 1. The non-normalised value of d index for *Picobia lamprotornis* (d = 1.96), *Picobia malayi* (d = 1.67), *Picobia sturni* (d = 1.61), and *Picobia indonesiana* (d = 1.14), *Picobia wisniewskii* (d = 1.86). We observed very high modularity in our network (likelihood = 0.77), within which we identified four modules, each containing hosts ranging from three to eight (Figure 4).

## 4. Discussion

In our studies, we found that, among the representatives of the subfamily Picobiinae, starlings are parasitized exclusively by those of the genus *Picobia*. This genus belongs to the *Picobia*-generic group and is the most species-rich genus in the family Picobiinae, with 45 described species [9,15]. Most species of this genus are associated with birds of the order Passeriformes (41 species), while a smaller number of *Picobia* species have been recorded from non-passeriform birds, such as Piciformes (three species) and Bucerotiformes (one species). Due to the numerous associations of the genus *Picobia* with passerine hosts, discovering representatives of this genus on starlings is not unexpected. More interestingly, the distribution of different *Picobia* species parasitising birds of the family Sturnidae shows a highly congruent pattern with the phylogeny of their hosts [37,38,39,40,41,42,43].

The clade of jungle starlings from the Indo–Pacific Islands region is a deep branch in sturnid phylogeny [16]. The recent distribution of these ancestral lineages indicates that the evolutionary origin of Sturnidae is possibly in Southeast Asia, perhaps towards the Southwestern Pacific [44]. Infested members of this host lineage are parasitised by *Picobia malayi* (on *Gracula*, *Strepsocitta,* and *Sarcops*) and *P. indonesiana* (on *Enodes*, *Mino,* and species of diverse island radiation of genus *Aplonis*). The main group of starlings, which is widespread in savanna and woodland regions in the Old World, is divided into crown radiations of Eurasian Savanna Starlings and African Starlings. All members of the Eurasian clade in our dataset, e.g., *Sturnus*, *Spodiopsar*, and *Acridotheres*, but also *Creatophora* (whose native range is in East and South Africa), are parasitised by *Picobia sturni*. Finally, the lineage of African Starlings, sister to Eurasian Savannah Starlings, was split into two distinct lineages [16]. The clade of African Red-winged Starlings of genus *Onychognathus* is parasitised by *Picobia wisniewskii*, while African Glossy Starlings (represented by *Lamprotornis*, *Poeoptera* in our data) are parasitised by *Picobia lapmrotornis*. It can be assumed that representatives of individual *Picobia* species were already present in the ancestors of specific clades of Sturnidae.

Currently, most picobiine mites are treated as monoxenous parasites (56% of all species), inhabiting only one host species. Oligoxenous, or mesostenoxenous species, which live on hosts within a single genus or family, comprise 18% and 22% of the picobiine species, respectively. Only a small fraction (4%) is metastenoxenous, associating with hosts across multiple families but restricted to a single avian order [8]. To date, we do not have records of mite species switching between representatives of different bird orders. The mite species associated with starlings can be classified as narrow oligoxenous parasites, such as *P. wisniewskii*, which is found on birds of the genus *Onychognathus*, and mesostenoxenous parasites, which include all other mite species, i.e., *P. indonesiana*, *P. malayi*, *P. sturni*, and *P. lamprotornis*, inhabiting birds from different genera. However, all of these species are specific to a particular phylogenetic clade of Sturnidae.

The Picobiinae–Sturnidae bipartite network (Figure 3) exhibited a low connectance value (Con = 0.25). This means that only 25% of all possible connections between picobiines and starlings are observed. This result is also confirmed by a C-score = 1, indicating that picobiines associated with this group of birds exhibit complete non-co-occurrence. Consequently, it is not surprising that the H_2_′ specialisation index is also 1, showing that picobiines are highly specialised and do not interact with each other. The temperature of nestedness, 54.49, indicates the moderate internal organisation of the network. The normalised specialisation level (d′) (maximum functional diversity) in all cases was 1, meaning that each of these parasite species has unique ways of interacting with hosts in the network. However, the non-normalised value of this index, “d”, was highest for *Picobia wisniewskii* (d = 1.86) and *Picobia lamprotornis* (d = 1.79), indicating that these species have the greatest diversity in host interactions compared to other species within this taxon (*P. malayi* d = 1.51, *P. sturni* d = 1.44, and *P. indonesiana* d = 0.97). As a result, we observed very high modularity in our network, identifying five distinct modules (Figure 4), each comprising hosts ranging from two to eight.

The present studies on the prevalence of infestation show a relatively high rate, with few species where prevalence is below 20% (Table 1). This rate can be correlated with the highly social behaviour of the family Sturnidae, cooperative breeding, and the fact that most species are cavity breeders [16]. It is worth noting that many of the studies of quill mites were conducted on museum skins in various collections. In such circumstances, whole quill mite populations remain enclosed in the calamus cavity of the feather quill, where they can be investigated even after many decades. Thus, one could ask—is it possible to consider such encapsulated population studies comparable with field studies performed on wild host populations? In this study, conducted entirely in museum avian collections, we found a prevalence of *Sturnus vulgaris* of 21.2% (all seasons pooled, different countries in the Palaearctic, see Table 3). Skoracki et al. [36] found in a study of *S. vulgaris* conducted in Poland (during the spring season) that the prevalence was 28.98%. These values do not differ significantly from each other (Fisher exact test, two-tailed *p* = 0.29), suggesting that population studies of quill mites performed in museum collections can be comparable to investigations in the field.

## 5. Conclusions

This study reveals the highly specialised and phylogenetically congruent nature of the interactions between quill mites of the subfamily Picobiinae and their starling hosts. We identified five species of quill mites parasitising 24 starling species, highlighting the narrow host specificity and significant specialisation within the Picobiinae–Sturnidae bipartite network. The low connectance value and high modularity of the network underscore the distinct and specialised nature of these host–parasite relationships. Our findings show that *Picobia* species exhibit a distribution pattern congruent with the phylogeny of their starling hosts, supporting the idea of co-evolution between mites and starlings. The high prevalence of mites is influenced by the social and breeding behaviours of starlings, which facilitate parasite transfers. Overall, this study enhances our understanding of the ecological and evolutionary dynamics of quill mites and starlings, emphasising the importance of specialisation and phylogenetic congruence in shaping these intricate relationships. Future research could further explore genetic mechanisms underlying host specificity and examine how environmental changes impact these specialised interactions.

## Figures and Tables

**Figure 3 animals-14-02517-f003:**
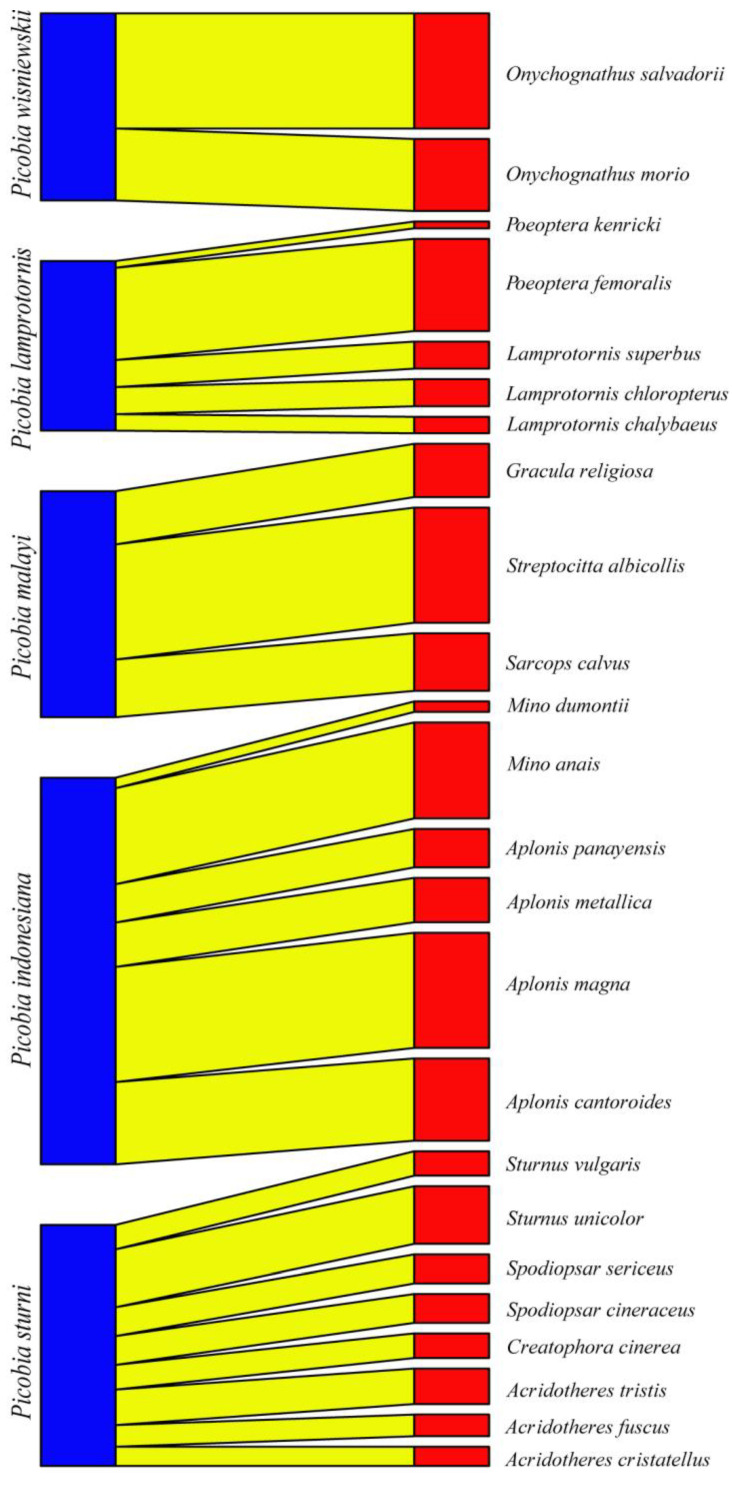
Bipartite network graph of interactions between Picobiine mites (**left**) and their Starling hosts (**right**).

**Figure 4 animals-14-02517-f004:**
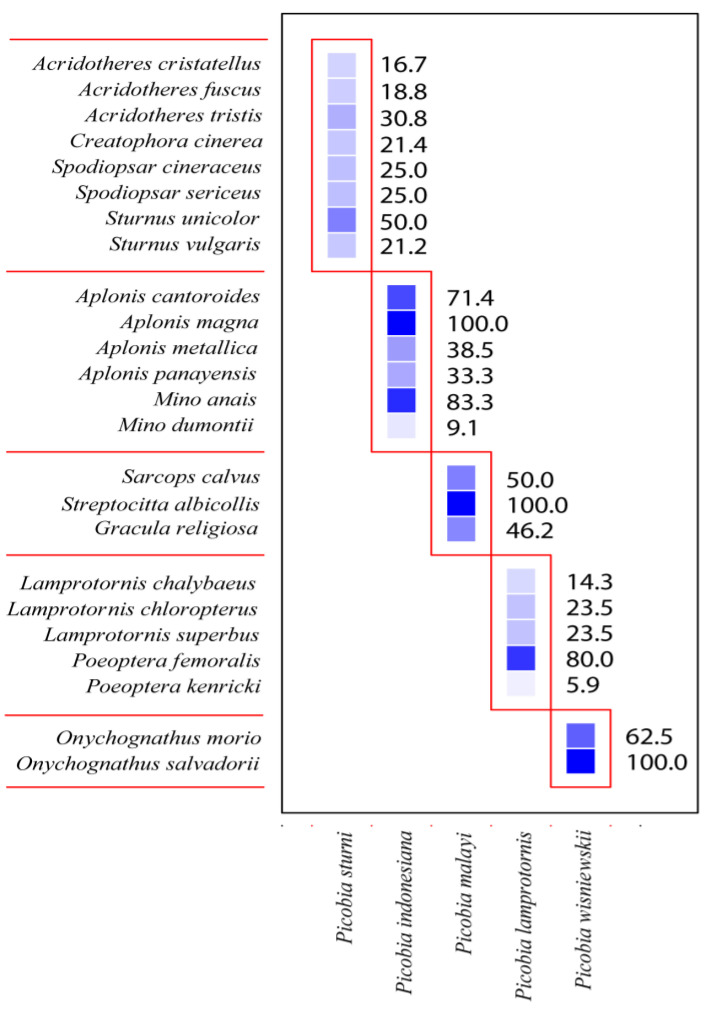
Modules of the Picobiine mites—Starlings communities. The intensity of the colour of each square indicates the strength of the interaction between particular parasite species (horizontal axis) and their host species (vertical axis).

**Table 1 animals-14-02517-t001:** Quill mite species of the subfamily Picobiinae (fam. Syringophilidae) parasitising starlings. (*)—type host species. Abbreviations for zoogeographical regions: Afro—Afrotropical, Orie—Oriental, Ocea—Oceanian, Pala—Palaearctic.

Mite Species	Host Species	Locality	References
*Picobia indonesiana* Skoracki and Glowska, 2008	*Aplonis panayensis* (Scopoli) *	Orie: Indonesia, India	[34]; current study
	*Aplonis metallica* (Temminck)	Orie: Indonesia, Ocea: Papua New Guinea	current study
	*Aplonis magna* (Schlegel)	Orie: Indonesia	current study
	*Aplonis cantoroides* (Gray)	Orie: Indonesia	current study
	*Enodes erythrophris* (Temminck)	Orie: Indonesia	[34]
	*Mino dumontii* (Lesson)	Orie: Indonesia, Ocea: Papua New Guinea	[34]; current study
	*Mino anais* (Lesson)	Ocea: Papua New Guinea	current study
*Picobia lamprotornis* Klimovičová, Skoracki, Wamiti and Hromada, 2014	*Lamprotornis superbus* Rüppell *	Afro: Kenya	[19,21]
	*Lamprotornis chalybaeus* Hemprich and Ehrenberg	Afro: Tanzania, Kenya	[21]
	*Lamprotornis chloropterus* Swainson	Afro: Tanzania	[21]
	*Poeoptera femoralis* (Richmond)	Afro: Tanzania	[35]
	*Poeoptera kenricki* Shelley	Afro: Tanzania	[35]
*Picobia malayi* Patan and Skoracki sp. n.	*Gracula religiosa* Linnaeus *	Orie: Indonesia	current study
	*Streptocitta albicollis* (Vieillot)	Orie: Indonesia	current study
	*Sarcops calvus* (Linnaeus)	Orie: Philippines	current study
*Picobia sturni* Skoracki, Bochkov and Wauthy, 2004	*Sturnus vulgaris* Linnaeus *	Pala: Poland, Moldova, Slovakia, Iceland, Kazakhstan, Kyrgyzstan, Uzbekistan, China	[11,20,36]; current study
	*Sturnus unicolor* Temminck	Pala: Italy, Spain, Morocco	[8]; current study
	*Spodiopsar cineraceus* (Temminck)	Pala: China, Japan	[11]; current study
	*Spodiopsar sericeus* (Gmelin)	Pala: China	current study
	*Acridotheres cristatellus* (Linnaeus)	Pala: China	current study
	*Acridotheres fuscus* (Wagler)	Pala: Nepal	current study
	*Acridotheres tristis* (Linnaeus)	Orie: India	current study
	*Creatophora cinerea* (Meuschen)	Afro: Kenya, Tanzania	[35]
*Picobia wisniewskii* Patan, Skoracki and Marcisova, 2024	*Onychognathus morio* (Linnaeus) *	Afro: Tanzania	[35]
	*Onychognathus salvadorii* (Sharpe)	Afro: Ethiopia	current study

**Table 2 animals-14-02517-t002:** Host species infested by quill mites with the index of prevalence (IP) and 95% confidence interval (CI, Sterne’s method).

Host Species	No. Examined	No. Infested	Prevalence (CI)	Mite Species
*Acridotheres cristatellus*	6	1	16.7% (0.9–58.9)	*Picobia sturni*
*Acridotheres fuscus*	16	3	18.8% (5.3–43.6)	*Picobia sturni*
*Acridotheres tristis*	13	4	30.8% (11.3–58.7)	*Picobia sturni*
*Aplonis cantoroides*	7	5	71.4% (34.1–94.7)	*Picobia indonesiana*
*Aplonis magna*	1	1	100% (5–100)	*Picobia indonesiana*
*Aplonis metallica*	26	10	38.5% (21.2–57.8)	*Picobia indonesiana*
*Aplonis panayensis*	18	6	33.3% (15.6–58.6)	*Picobia indonesiana*
*Creatophora cinerea*	14	3	21.4% (6.1–50)	*Picobia sturni*
*Gracula religiosa*	13	6	46.2% (22.4–74)	*Picobia malayi*
*Lamprotornis chalybaeus*	14	2	14.3% (2.6–42.6)	*Picobia lamprotornis*
*Lamprotornis chloropterus*	17	4	23.5% (8.5–48.9)	*Picobia lamprotornis*
*Lamprotornis superbus*	17	4	23.5% (8.5–48.9)	*Picobia lamprotornis*
*Mino anais*	6	5	83.3% (41.1–99.2)	*Picobia indonesiana*
*Mino dumontii*	11	1	9.1% (0.5–40.5)	*Picobia indonesiana*
*Onychognathus morio*	8	5	62.5% (28.9–88.9)	*Picobia wisniewskii*
*Onychognathus salvadorii*	1	1	100% (5–100)	*Picobia wisniewskii*
*Poeoptera femoralis*	5	4	80% (34.3–99)	*Picobia lamprotornis*
*Poeoptera kenricki*	17	1	5.9% (0.3–28.7)	*Picobia lamprotornis*
*Sarcops calvus*	4	2	50% (9.8–90.2)	*Picobia malayi* sp. n.
*Spodiopsar cineraceus*	16	4	25% (9–50)	*Picobia sturni*
*Spodiopsar sericeus*	8	2	25% (4.6–63.5)	*Picobia sturni*
*Streptocitta albicollis*	1	1	100% (5–100)	*Picobia malayi* sp. n.
*Sturnus unicolor*	12	6	50% (24.3–75.7)	*Picobia sturni*
*Sturnus vulgaris*	104	22	21.2% (14.3–30.2)	*Picobia sturni*

**Table 3 animals-14-02517-t003:** Uninfested starling species.

Starling Species	N	Starling Species	N	Starling Species	N
*Acridotheres melanopterus*	1	*Aplonis minor*	6	*Aplonis opaca*	2
*Hartlaubius auratus*	1	*Hylopsar cupreocauda*	1	*Hylopsar purpureiceps*	1
*Lamprotornis australis*	1	*Lamprotornis caudatus*	2	*Lamprotornis fischeri*	5
*Lamprotornis nitens*	6	*Lamprotornis purpureus*	3	*Lamprotornis purpuroptera*	5
*Lamprotornis regius*	3	*Lamprotornis splendidus*	4	*Onychognathus tristramii*	1
*Poeoptera sharpii*	11	*Poeoptera stuhlmanni*	1	*Saroglossa spilopterus*	2
*Speculipastor bicolor*	1	*Sturnia sinensis*	2		

## Data Availability

All necessary data are available in the text and the Appendix A for this article.

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
