# Peer review of "Diversity and Interactions between Picobiine Mites and Starlings"

_animals, 2024, doi:10.3390/ani14172517_

Round 1

Reviewer 1 Report

Comments and Suggestions for Authors

In the MS the authors reviewed the data on the quill mites associated with starlings. They focused on the mites of the genus Picobia, described a new species of this genus and provide new data on the distribution of these mites. The authors performed statistical analyses in order to investigate the diversity, interactions, and level of specialization between mites and their bird hosts on a worldwide scale. In general, the MS is well written. It includes several good illustrations. The statistical analyses were conducted correctly. The MS needs minor revision. Some additional remarks are below.

Title (Diversity and interactions between Picobiine mites and StarLings) – the title does not reflect some important content of the study, e.g. description of the a species. Additionally, it is recommended to included several key words (e.g. “birds”, “parasites”, “quill or feather mites” etc.) and indication of geography (world diversity/ reagional diversity) in order to make the title more attractive to broader audience.

20: “We identified five species of quill mites infesting 24 species” --- please, specify geography. Do you mean the world fauna?

23: “specialised nature” – please, revise this world combination

27,28 vs 23,24: please compare these two sentences and revise them to avoid repetition

“The network structure demonstrated four distinct modules, emphasising the specialised nature of these associations.”

“This comprehensive network analysis provides new insights into the ecological dynamics of host-parasite interactions, highlighting the specialised nature of these relationships”.

31: “In this paper,” --- redundant

Abstract: please, indicate geography of the study

39: “mesostenoxenous nature” --- please, revise this world combination

40: “The distribution of Picobia species…” --- please consider adding “among sturnids” or something like that after “Picobia species”

41: “The findings suggest that the social and breeding behaviours of starlings influence quite a high prevalence.” --- this statement does not follow from the previous sentences of the Abstract. Could you slightly revise it and provide a better connection with other sentences in the Abstract?

52: please, provide brief systematics of the host in brackets

53: “long and needle-like” --- either “long” or “and” is redundant

92: “multiple objectives” --- according to the text, the study has two objectives (“first… second…”)

2.1. Host sampling --- please indicate the number of the investigated birds of each species in this section, e.g. n=…

109: Hoyer’s medium – please, specify which medium exactly did you use. See here: Upton 1993 (https://doi.org/10.1017/S0007485300034763)

2.4. Statistical analyses --- please, describe your matrix and explain which data was included/excluded. Otherwise, it is not clear which data exactly was analyzed and what is the origin of the analyzed data.

167: chapter --- please, check this word, may be “section” is more adequate

Fig.1 could you make the outlines of the mite body a bit more distinct using bold line?

Fig.1 could you provided high quality microphotographs of the 5 studied picobiins?

Fig.3. This figure seems to be deformed (the font of the text is confusing), probably, you could revise it

307: “In our studies, we found that starlings are parasitised exclusively by representatives of the genus Picobia.” --- please, be more precise: there are many parasites, you probably mean only quill mites

308-311: “This genus belongs to the Picobia-generic group and is the most species-rich genus in the family Picobiinae, with 45 described species [9, 15]. Most species of this genus are associated with birds of the order Passeriformes (41 species), while a smaller number of Picobia species have been recorded from non-passeriform birds, such as Piciformes (three species) and Bucerotiformes (one species)” --- this data on Picobia should be given in the Introduction, otherwise, after having read the Introduction some lack of the information on the Picobia is evident.

315: “a highly congruent pattern with the phylogeny of their hosts” --- could you please provide an illustration to visualize this statement? You could map Picobia spp on the strunid phylogeny and make your discussion much more attractive and easier to understand for readers

320: by --- non italic

359-373: the hypothesis on the correlation between bird behavior and mite distribution needs better explanation as well as the hypothesis on the bird-mite coevolution

Comments on the Quality of English Language

minor

Author Response

Dear Reviewer,

Thank you very much for your time and the valuable comments you provided on the first version of our manuscript. Your insights have been incredibly helpful in improving the quality and clarity of our work. We greatly appreciate your thorough review and constructive feedback.

Best regards,

Authors

PS. Please find attached our responses to the comments.

Reviewer 2 Report

Comments and Suggestions for Authors

Dear Authors,

this study deals with the diversity of quill mites of the subfamily Picobiinae (Syringophilidae) parasitasing starling birds and the investigation of the host-parasite interactions. The study revealed the presence of 5 species of the genus Picobia parasitising 24 species of starlings, one of them described as new to science. The results showed narrow host specificity and significant specialisation between these mites and starling birds. 

The manuscript is well-written, presenting interesting findings. The description of the new species is very detailed and the drawings are of good quality. The methods used for exploring the host-parasites interactions are adequate. The references cited are appropriate and the conclusions are supported by the results. 

For the aformentioned reasons I recommend the acceptance of the manuscript for publication in its present form. 

Sincerely

Author Response

We would like to sincerely thank the Reviewer for the positive review and for taking the time to thoroughly read our manuscript. 

Sincerely,

Authors

Reviewer 3 Report

Comments and Suggestions for Authors

Review

of the article “Diversity and interactions between Picobiine mites and Starlings”

by the authors Bozena Sikora, Jakub Z. Kosicki, Milena Patan, Iva Marcisova, Martin Hromada and Maciej Skoracki

Feather mites that parasitize birds of the starling family (Sturnidae) have not yet been sufficiently studied. The subfamily Picobiinae (Acariformes: Syringophilidae) comprises obligate and permanent parasites of birds found exclusively in the quills of contour feathers. In this paper, the authors studied the associations of picobiine mites with birds of the Sturnidae family (Aves: Passeriformes). Among the 414 examined bird individuals belonging to 44 species (35.2% of all sturnids), 103 individuals from 24 species (54.5% of examined species) were parasitised by quill mites. Mite diversity was represented by five species, including one newly described species by the authors, Picobia malayi Patan and Skoracki sp. n.

The research presented continues large-scale studies focusing on quill mites across particular families within the order Passeriformes. The aims of this study are to first describe a new species of the genus Picobia and secondly to detail the diversity, interactions and level of specialization between syringophilid ectoparasites and their starling hosts globally. The mite material used in this paper was collected from dry bird skins, mainly housed in the ornithological collection of the Bavarian State Collection of Zoology (SNSB-ZSM) in Munich, Germany, as well as in other museum collections.

Statistical processing of the obtained data was carried out. To describe the patterns in the studied two-way host-parasite ecological network, the "two-way" package for R was used. Descriptive statistics were calculated using Quantitative Parasitology v.3.0 on the Web.

In this study, the authors identified five species of mites of the subfamily Picobiinae parasitizing 24 species of starlings. A list of mite species together with their hosts and distributions is presented, including new records. The authors compiled a key to all picobiine species associated with birds of the family Sturnidae. A description of the new species Picobia malayi Patan and Skoracki sp. n. is presented.

The distribution of Picobia species was shown to correspond with the phylogeny of their starling hosts, with different mites parasitizing specific starling clades. The findings suggest that the social and breeding behaviours of starlings influence quite a high prevalence.

The “Abstract” outlines the main points of the conducted research. In the “Introduction” section, the need for the study is substantiated. The “Introduction” includes a complete review of the literature sources on the problem. The references cited are current. The “Materials and Methods” section is written in sufficient detail and clearly.

The “Results” section is written in sufficient detail and clearly; three tables and four figures illustrate the results obtained. The “Discussion” section is based on the presented results and contains an analysis of literature data. The “Conclusion” section is based on the results and conclusions drawn from the discussion of the findings. A sufficient list of references is provided. 44 sources were analyzed. No excessive self-citation.

The list with full data of the collected mite material is presented in Supplement 1.

In general, after minor editorial corrections, the manuscript can be published in the “Animals” journal, the “Ecology and Conservation” Section, of Special Issue “Diversity and Interactions Between Mites and Vertebrates”.

Author Response

(The authors gave the same response as above.)
